# Subjective value and decision entropy are jointly encoded by aligned gradients across the human brain

Sebastian Bobadilla-Suarez [1 ✉], Olivia Guest [1,2] & Bradley C. Love [1,3]

Recent work has considered the relationship between value and confidence in both behavioural and neural representation. Here we evaluated whether the brain organises value and confidence signals in a systematic fashion that reflects the overall desirability of options. If so, regions that respond to either increases or decreases in both value and confidence should be widespread. We strongly confirmed these predictions through a model-based fMRI analysis of a mixed gambles task that assessed subjective value (SV) and inverse decision entropy (iDE), which is related to confidence. Purported value areas more strongly signalled iDE than SV, underscoring how intertwined value and confidence are. A gradient tied to the desirability of actions transitioned from positive SV and iDE in ventromedial prefrontal cortex to negative SV and iDE in dorsal medial prefrontal cortex. This alignment of SV and iDE signals could support retrospective evaluation to guide learning and subsequent decisions.

[1] Department of Experimental Psychology, University College London, 26 Bedford Way, London WC1H 0AP, UK. [2] Research Centre on Interactive Media, Smart Systems and Emerging Technologies — RISE, Nicosia, Cyprus. [3] The Alan Turing Institute, 96 Euston Road, London NW1 2DB, UK. ✉email: sebastian.suarez.12@ucl.ac.uk

Subjective value (SV) and confidence are closely linked concepts. For instance, people tend to be highly confident in accepting a high-value option (e.g., their dream job). Similarly, they are confident when rejecting a low-value option (e.g., spoiled milk). For middling values, people will be uncertain of what choice to make and confidence will be low.

Shannon entropy is a well-formulated measure of uncertainty[1] that is well suited for examining confidence. So that it positively aligns with confidence, we consider the inverse of the entropy associated with a person's decision, which we refer to as inverse decision entropy (iDE). Shannon entropy characterises the uncertainty for a probability distribution in terms of the expected self-information, which can be calculated as the sum of the probability of each state times its log probability. In the case of the binary decisions considered here, the probability distribution is simply a binomial. In other words, the relationship between SV and iDE can be described by a simple mathematical function that transforms SV into the probability of accepting an option (Fig. 1b[2–5]) and this probability in turn can be transformed into iDE. Although closely related conceptually, SV and iDE need not correlate (Fig. 1b). Indeed, all combinations of low and high values are possible for SV and iDE (see Fig. 1c).

Research in value-based decision making has considered measures related to confidence, such as risk, decision uncertainty, or the subjective probability of being correct (i.e., confidence[4,6,7]). For example, decision confidence can be operationalised as a quadratic transform of SV (i.e., with an inverted-U relation to value[2–5,8]) and a sigmoidal relation with choice probability (see Fig. 1b), estimated from a cognitive model[6,9,10], or elicited as a subjective rating[6,11,12]. Algorithmic proposals link confidence to evidence accumulation in value-based decision making[6,13,14].

One interesting question is how these value and confidence signals relate. One idea is that the evidence accumulation with respect to a value comparison process is performed in vmPFC and the confidence in this decision is explicitly represented in rostrolateral PFC, enabling verbal reports of confidence[6,11]. In line with the notion that SV and confidence are interlinked, confidence signals have been found more dorsally than SV on the medial surface of prefrontal cortex[4,6,12]. Although confidence or decision entropy can accompany SV computations for many of the mentioned regions[6,13,15], it is not yet clear whether areas that encode value also encode confidence and vice versa. At this juncture, rather than focusing on their localisation, we suggest mapping the relationship between confidence and value throughout the brain with a focus on gradients[16].

Lebreton et al.[4] suggested that representations of value and confidence are combined into a single quantity (i.e., in vmPFC). Similarly, Gherman and Philiastides[17] also found evidence for decision confidence signals in vmPFC but for a perceptual discrimination task. Intuitively, confidence can be seen as having value in-and-of-itself that inflates the basic value signal. Although by definition the immediate decision is driven by value, a more encompassing evaluation of a decision may involve confidence, which could shape future behaviour[18]. We find this basic account appealing, but incomplete. Lebreton et al.[4] focused on the case of positive coding of value and confidence in vmPFC. If value and confidence signals are truly intertwined, then there should also be regions that code the converse; negative coefficients for value and confidence, which is equivalent to increased activity for low confidence and negative value. Furthermore, evaluating uncertainty negatively is consistent with studies of risk aversion[7,19,20] and related to anxiety disorders or depression[21].

Moreover, one might expect cortical maps that smoothly vary, in a gradient-like manner[16,22], from positive options (high value,

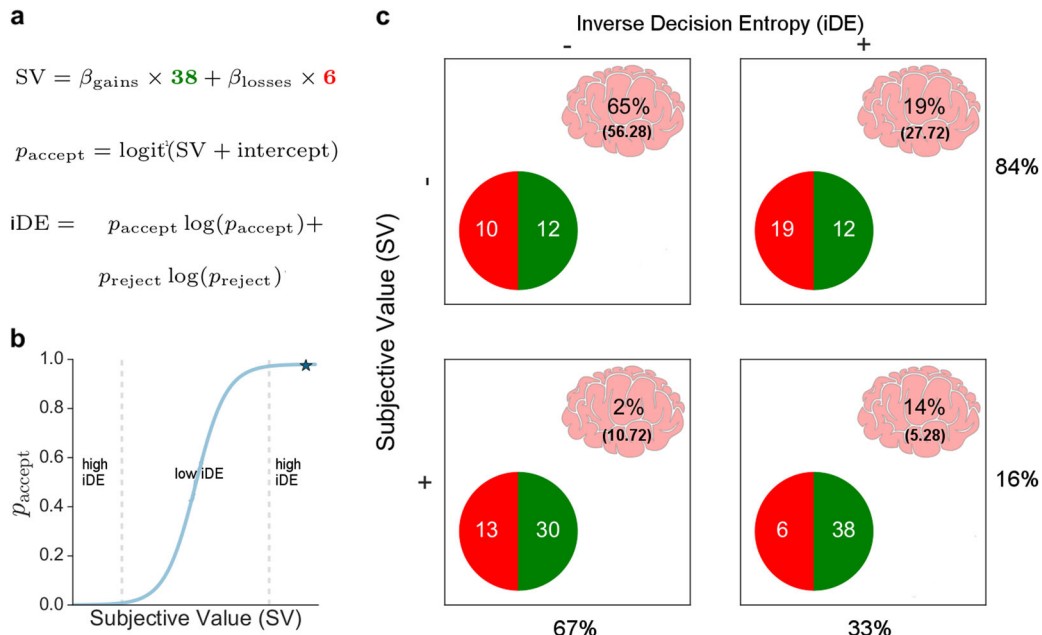

**Fig. 1 Behavioural analysis and voxel distribution.** Three equations (**a**) describe the behavioural model in which subjective value (SV) is a weighted combination of gains and losses, $p_{accept}$ is the probability of accepting a gamble, and inverse decision entropy (iDE) is the (negative) Shannon entropy of $p_{accept}$ and its complement $p_{reject}$. **b** $p_{accept}$ is a function of SV. High values of iDE arise from extreme values of SV, whereas iDE is low for middling values of SV in which $p_{accept}$ is close to 0.5. **c** The 2x2 table shows all positive and negative combinations of SV and iDE. In each cell, the percentage of voxels (whole brain) that show that specific combination of SV and iDE effects is shown along with the expected percentage in parentheses according to the null hypothesis that SV value and iDE are independent. The results indicate SV and iDE tend to both be either positive or negative. The marginals for the rows and columns are also shown. The gambles in each cell are meant to represent different combinations of high and low SV and iDE for a typical participant that presents loss aversion (e.g., $\beta_{losses} \approx -2\beta_{gains}$).

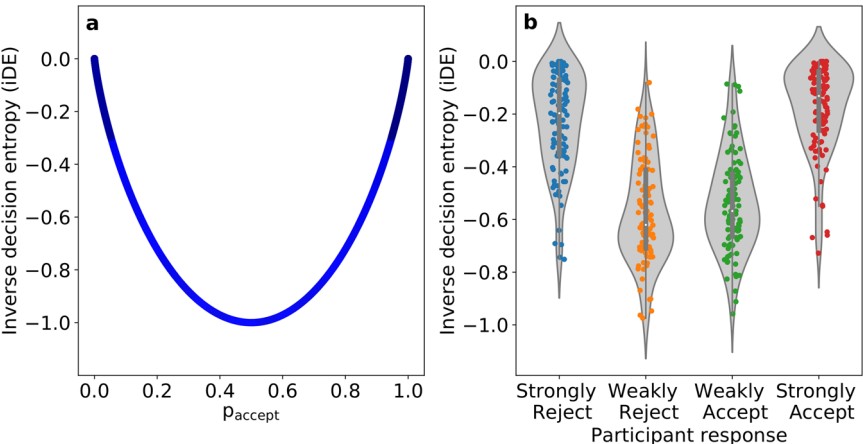

**Fig. 2 Inverse decision entropy (iDE) and response probabilities. a** The U-shaped relationship between $p_{accept}$ and iDE is shown for over twenty-four thousand choices across all participants ($N = 104$). Confidence is highest for low-value gambles that have a low probability of acceptance and high-value gambles that have a high probability of acceptance. The transparency in the plot reflects the density of observations in the empirical data along the $p_{accept}$ horizontal axis. **b** A plot of iDE as a function of the four possible responses: Strongly Reject, Weakly Reject, Weakly Accept, Strongly Accept. Each dot is a participant's mean for that response type and the grey conveys the density across participants. Although our cognitive model was fit to the binary distinction of accept vs. reject, it successfully generalised by showing sensitivity to the weakly vs. strong distinction for which it was not fit.

high confidence) to negative options (low value, low confidence). According to this account, the distribution of voxels across the brain that code for value and confidence will be highly non-accidental: (1) voxels that code for value should also code for confidence; and vice versa, (2) most voxels sensitive to value and confidence should either code for negative value and low confidence or positive value and high confidence. Thus, this study characterises the joint neural coding of value and confidence on the medial surface of the human brain.

To foreshadow our results, these predictions were confirmed. We observed a gradient (on the medial surface of PFC) that tracked both value and iDE (i.e., confidence) in a principled way. Thus, what we find are representations geared towards evaluating actions; a decision map that is activated from low confidence (low iDE) and low value in dorsomedial prefrontal cortex (dmPFC) to high value and high confidence (high iDE) in vmPFC. We also found that positive/positive and negative/negative relationship between value and confidence held in voxels throughout the brain. The contribution of this study to decision neuroscience is twofold. First, the joint coding of value and confidence, previously proposed by Lebreton et al.[4] for positive value and high confidence, is extended to consider the converse case and the distribution of voxels jointly coding value and confidence. Furthermore, our results suggest that medial surface activity is best described by large-scale maps for decision and action related computations. Our results indicate that SV and confidence gradients in the brain are aligned in a manner that reflects the overall desirability of a decision, which could be useful in retrospective evaluation of a decision.

To specify this neural link between decision entropy and SV, we used fMRI data from the Neuroimaging Analysis Replication and Prediction Study [NARPS[23,24]]. With a considerably large sample size ($N = 104$, after exclusion), we tested the different contributions of SV and decision entropy to the blood oxygen level dependent (BOLD) signal. Sample sizes as large as these are uncommon for neuroeconomic experiments, which makes this dataset well suited to answering how value and confidence are related in the brain at large. We pitted iDE and SV against each other with a focus on a whole-brain corrected analysis of three canonical value areas: nucleus accumbens (NA), vmPFC, and the amygdala. These regions of interest (ROI) were pre-selected in the original NARPS study (see Original NARPS ex-ante hypotheses

in the SI) which focused on the analysis of gains and losses but not confidence. The task was a mixed gambling task where participants either accepted or rejected each gamble.

## Results
The results are based on data collected by the NARPS team[23,24]. After applying exclusion criteria (see "Methods"), data from 104 participants from the mixed gambles task were analysed. In the scanner, they were asked to accept or reject prospects with a 50% chance of gaining or losing a certain amount of money.

Decision weights for gains and losses were estimated for each participant by logistic regression on the decision to accept or reject the gamble. This approach models how biased a participant is when accepting or rejecting a given gamble, based on properties of that gamble. The logistic regression models the participants' probability, $p_{accept}$, of accepting a gamble on a given trial as

$$p_{accept} = logit^{-1}(\beta_{gains} \times gains + \beta_{losses} \times losses + intercept). \quad (1)$$

Using our model we computed the SV, which is how much a participant values the current gamble, and the iDE, which is how certain a participant is about accepting or rejecting the current gamble. SV for a specific trial was computed using the estimated beta coefficients $\beta$ for gains ($\beta_{gains}$) and losses ($\beta_{losses}$) as

$$SV = \beta_{gains} \times gains + \beta_{losses} \times losses. \quad (2)$$

From $p_{accept}$, we calculate decision (Shannon) entropy as

$$DE = -[p_{accept} \times \log_2(p_{accept}) + p_{reject} \times \log_2(p_{reject})], \quad (3)$$

where $p_{reject}$ is $1 - p_{accept}$. Finally, iDE is simply negative DE. Although simple, this model captures individual differences in both behaviour and brain response (see Supplementary Fig. 1 and Behavioural model in the Supplemental Information, SI). For example, estimated behavioural loss aversion for a participant, $\beta_{losses}/\beta_{gains}$, tracked the ratio of negative and positive SV voxels (see Supplementary Fig. 4 and Loss aversion in the brain in the SI).

As can be seen in Fig. 2a, iDE has a quadratic relation to $p_{accept}$ with a significant (above zero) mean Spearman correlation of 0.16 (s.d. = 0.568, $t(103) = 2.88$, $p = 0.005$) across participants. The density of observations for $p_{accept}$—estimated with one thousand

bins for over twenty-four thousand choices across participants—is biased towards towards the upper and lower bounds (i.e., $p_{accept} = 1$ and $p_{accept} = 0$, respectively). Likewise, iDE shares a quadratic relation with SV (see Supplementary Fig. 3) presenting a significant (above zero) mean Spearman correlation of 0.16 (s.d. = 0.568, $t(103) = 2.88$, $p = 0.005$), which follows from the high Spearman correlation between SV and $p_{accept}$ (mean = 0.99, s.d. = 0.0004, one sample $t$-test above zero: $t(103) = 28687.295$, $p < 0.001$).

Our iDE measure of confidence closely tracks other measures in the literature. For example, iDE positively correlates with confidence ratings provided by participants in a behavioural study ($n = 28$[18], see Validation of iDE in the SI) of value-based decision making with a Spearman correlation of 0.45 (s.d. = 0.171, $t(27) = 13.61$, $p < 0.001$). In that study, iDE was closely related to the authors' preferred definition of confidence, namely the subjective probability of being correct (above zero mean Spearman correlation of 0.89, s.d. = 0.114, $t(27) = 40.65$, $p < 0.001$). This measure of confidence and iDE also tracked one another in the current study using the NARPS data (above zero mean Spearman correlation of 0.96, s.d. = 0.046, $t(103) = 210.426$, $p < 0.001$). These relations hold for alternative definitions of value as well (see Supplementary Fig. 2 and Validation of iDE in the SI).

To evaluate the robustness of iDE, we considered how it varied for strongly vs. weakly accepts and rejects. Although we modelled iDE based on the accept vs. reject binary distinction, participants had four responses available to them. Even though our model fit was not informed by the strongly vs. weakly distinction, one would hope that iDE would be lower for the weakly accept and reject responses than for the strongly accept and reject responses. Indeed, as shown in Fig. 2b, this relation held. The lower confidence responses (Weakly Reject and Weakly Accept) showed lower iDE (mean Weak iDE of $-0.55$, s.d. = 0.189) than the Strongly Accept and the Strongly Reject options (mean Strong iDE of $-0.20$, s.d. = 0.177, significantly higher than Weak iDE: $t(406.61) = 19.31$, $p < 0.001$).

Both SV and iDE, estimated from behavioural, were used as parametric modulators in a general linear model (GLM) of the fMRI data. This model-based fMRI analysis answers three key questions: (1) How widespread are the effects (either positive or negative) of SV and iDE? (2) Which areas differentially respond to either iDE or SV? and (3) How do SV and iDE effects interrelate?

**Main effects of SV and iDE**. The answer to the first question is shown in the left side of Fig. 3. Overall, it is striking how widespread SV and iDE effects (both positive and negative) are. To foreshadow the results, although both SV and iDE signals are widespread, iDE is more pervasive. Areas that signal both SV and iDE tend to respond either positively and negatively for both measures with a positive cluster in vmPFC and a negative cluster occurring more dorsally.

Negative effects of SV and iDE were not observed in NA, amygdala or vmPFC. Though SV (purple colours, top row in Fig. 3) indeed presented a strong cluster of deactivation (150,923 voxels, $p < 0.001$) with a peak $Z$ statistic of 8.39 (coordinates in MNI152 space in millimetres: $x = -44$, $y = -27$, $z = 61$) in the left postcentral gyrus. Also in Fig. 3 (left column), iDE (dark pink colours) presents a cluster of negative activation in the cingulate gyrus (3438 voxels, $p < 0.001$, peak $Z = 5.86$). However, the largest cluster of negative activation for iDE (300,573 voxels, $p < 0.001$) shows a peak $Z$ statistic in the right supramarginal gyrus of 10.2 (coordinates in MNI152 space in millimetres: $x = 50$, $y = -39$, $z = 53$). For the conjunction analysis of negative

effects, the top left brain in Fig. 3 (light pink colours) presents clusters with peak activation in left postcentral gyrus (25,820 voxels, $p < 0.001$, peak $Z = 5.76$) and cingulate gyrus (14,195 voxels, $p < 0.001$, peak $Z = 4.93$), among others (see Supplementary Table 1 for a list of all main effects).

As for positive effects, SV (purple colours, bottom left of Fig. 3) presents a strong cluster of positive activation (17,326 voxels, $p < 0.001$) in the right NA with a peak $Z$ statistic of 5.44 (coordinates in MNI152 space in millimetres: $x = 13$, $y = 15$, $z = -10$). Notably, activation of vmPFC was strong and part of the same cluster as right NA, extending towards the frontal pole with $Z$ statistics ranging from ~2.5 to ~4. No positive activations of SV were observed in bilateral amygdala. Also in Fig. 3 (dark pink colours, middle column), iDE presents an enormous cluster of positive activation (515,033 voxels, $p < 0.001$) with a peak $Z$ statistic in right vmPFC of 8.75 (coordinates in MNI152 space in millimetres: $x = 6$, $y = 56$, $z = -20$). This cluster extends towards bilateral NA and bilateral amygdala and is bigger than any cluster of activation found for SV, by far. For the conjunction analysis of positive effects (Fig. 3, light pink colours, middle brain in the bottom row), we found only one significant cluster with peak activation in vmPFC with activation extending into bilateral NA (14,732 voxels, $p < 0.001$, peak $Z = 5.02$, coordinates in MNI152: $x = 7$, $y = 51$, $z = -20$).

How widespread SV and iDE related activity is noteworthy. Furthermore, the alignment of negative effects (Fig. 3, top left) and positive effects (Fig. 3, middle column, bottom row) of both variables suggests a principled organisation for a decision-oriented map in mPFC.

Accordingly, SV and iDE effects were not as widespread with positive/negative or negative/positive pairings. Indeed, we found no cluster activations for the conjunction of positive SV with negative iDE (Fig. 3, light pink colours, bottom left). However, for the conjunction of negative SV and positive iDE (Fig. 3, light pink colours, middle column, top row), we found clusters with peak activation in the left and right supramarginal gyrus (respectively: 15,390 voxels, $p < 0.001$, peak $Z = 5.06$, and 8805 voxels, $p < 0.001$, peak $Z = 4.55$) as well as in the left postcentral gyrus, right lateral occipital cortex (LOC), and cingulate gyrus (see Supplementary Table 2 for more details on all conjunction clusters).

**Contrast of SV and iDE**. Our second question about preferential coding of SV or iDE is answered through the direct comparison of the effects of iDE and SV (Fig. 3, contrasts on the rightmost column). To avoid detecting stronger effects of one variable due to negative effects of the other, we performed a conjunction analysis of main effects with each contrast (see "Methods"). The main result is that iDE effects, both positive and negative, were stronger even in purported value areas.

As seen on the right-hand side of Fig. 3 (contrasts, bottom right), iDE had a larger overall positive effect when compared to SV. In accordance with the biggest iDE cluster observed in Fig. 3 (middle column), here we observed a cluster of 311,318 voxels ($p < 0.001$) with a mean $Z$ statistic of 3.2. Both vmPFC and bilateral amygdala were part of this cluster with $Z$ statistics close to the mean effect (within a tolerance of plus ~0.3 or minus ~0.7). For cerebral clusters where iDE showed a stronger negative effect than SV (Fig. 3, top right), these included: left and right frontal pole (respectively: 154059 voxels, $p < 0.001$, peak $Z = 3.54$, and 106,855 voxels, $p < 0.001$, peak $Z = 3.54$), left and right LOC (respectively: 7920 voxels, $p < 0.001$, peak $Z = 3.54$, and 10039 voxels, $p < 0.001$, peak $Z = 3.54$). The results did not show any clusters where SV had a significantly larger positive effect than iDE, which is striking for purported value areas. On the other

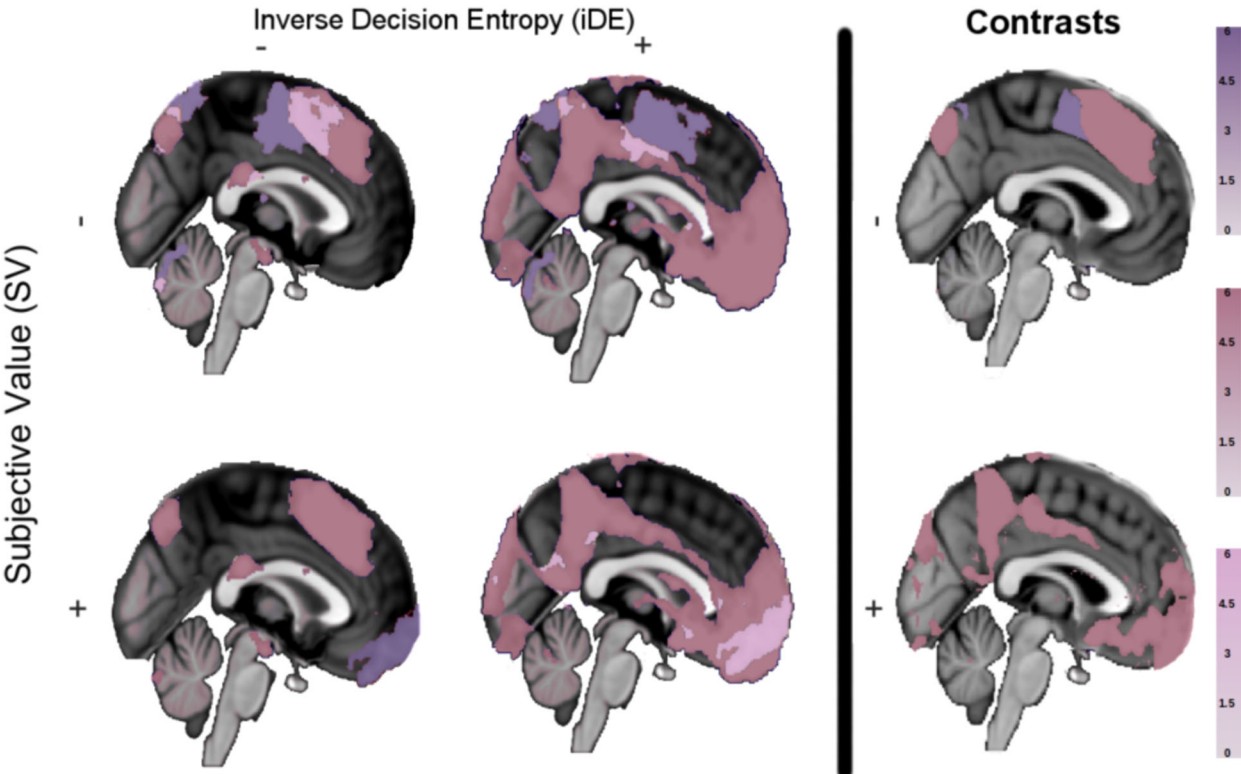

**Fig. 3 Main effects and contrasts in medial prefrontal cortex.** The first two columns on the left present significant activations (Z statistical maps) of subjective value (purple), inverse decision entropy (dark pink), and their conjunction (light pink) for a whole-brain corrected analysis conducted with FSL FEAT's FLAME 1 for different combinations of positive and negative main effects (2×2). The column on the right-hand side (i.e., contrasts of SV versus iDE) shows areas with stronger negative effects (top right) or stronger positive effects (lower right) of either subjective value or decision entropy.

hand, by far the biggest cluster where SV had a stronger negative effect than iDE (Fig. 3, top right) displayed peak activation in the left cingulate gyrus (29,900 voxels, $p < 0.001$, peak $Z = 3.54$). The low variance in the peak $Z$ statistics reported in this section was due to the nature of the test (see "Methods").

To summarise these results, iDE had a stronger effect in the amygdala bilaterally and vmPFC. No significant difference between SV and iDE was found in either left or right NA. Indeed, the contrast plots (Fig. 3, rightmost column) show that many traditional value areas are more responsive to entropy. More details on all clusters contrasting SV and iDE can be found in Supplementary Table 3 in the SI.

**Interdependence of SV and iDE.** Our final question concerns the relationship between SV and iDE. We predicted that these quantities would be intertwined in a particular way, namely that SV and iDE would collocate and match in terms of positivity and negativity. We confirmed these predictions in three ways.

First, in Fig. 1c, we present the different contingencies for the intersection of voxels where both variables have an effect in the whole brain (masked with task-active voxels), $\chi^2 = 25.59$, $p < 0.001$. This analysis found that voxels tend to either be both positive for SV and iDE or both negative. Figure 1c shows the expected and observed cell frequencies underlying this analysis. One observation is that there is also a strong effect for voxels to code negative values for both iDE and SV, which might relate to risk aversion. The relationship between iDE and SV was even stronger in three regions of interest (right NA, right amygdala, and frontal medial cortex - which includes vmPFC). Right NA had a 98% overlap of positive SV and iDE, whereas frontal medial cortex and right amygdala had 100% overlap.

Second, rather than dichotomise the data, we present the correlations of beta weights between SV and iDE for these same areas (Fig. 4). Frontal medial cortex shows the strongest correlation for these variables (Fig. 4e), $r = 0.823$, $p < 0.001$, and that the correlation remains positive at the whole-brain level (Fig. 4f), $r = 0.379$, $p < 0.001$. Both left NA (Fig. 4a), $r = 0.506$, $p < 0.001$, and right NA (Fig. 4b), $r = 0.488$, $p < 0.001$, show strong correlations between SV and iDE as well, followed by the right amygdala (Fig. 4d), $r = 0.281$, $p < 0.001$. The left amygdala (Fig. 4c) also shows an association but the effect is relatively small when compared to the other regions, $r = 0.141$, $p < 0.001$. These correlations involve only two statistical maps: one of SV and one of iDE. Each map was estimated from the same GLM across all participants with FSL's mixed effects model with outlier deweighting (FLAME 1, see "Methods"). For example, the dots in Fig. 4 represent activity in voxels of a Montreal Neurological Institute (MNI) brain template (averaged across participants, see "Methods"). The generalised interdependence between SV and iDE further supports the notion of a principled alignment between both measures. For completeness, we also correlated the voxelwise $Z$ statistics (which incorporated the voxel-specific variance across subjects; see Correlations of voxelwise $Z$ statistics between SV and iDE in the SI) and found the same pattern of results with the magnitude of the correlations slightly lower.

Third, the relation between SV and iDE formed smooth maps, as opposed to parcellations[16], that spanned large regions that were either positive or negative for both SV and iDE. For illustrative purposes, we present the beta weights (z-scored independently) for both variables viewed from a sagittal perspective of the medial cortex (Fig. 5). Notice that the areas that were positive or negative for SV (Fig. 5a) and iDE (Fig. 5b) tended to overlap such that the summation (Fig. 5c) reveals

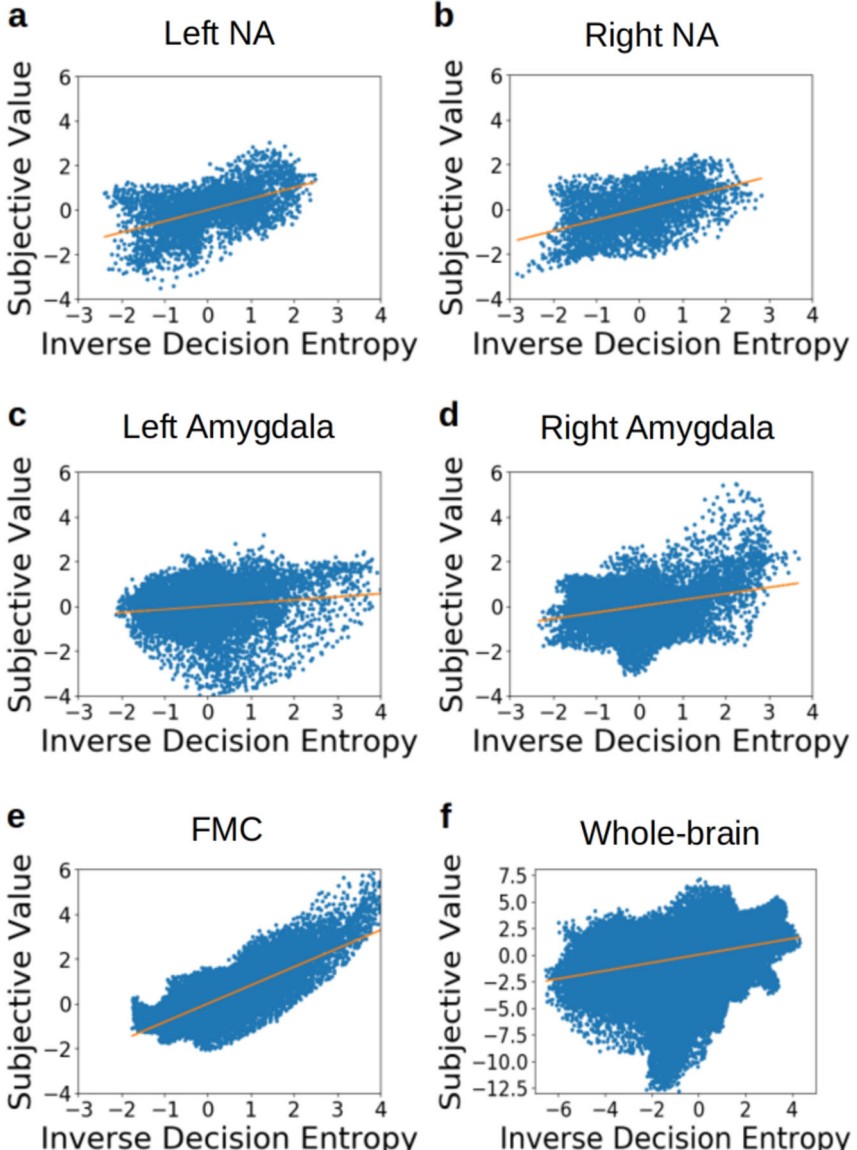

**Fig. 4 Links between subjective value (SV) and inverse decision entropy (iDE) across Regions of Interest (ROI).** SV and iDE positively correlate across voxels (**a**) left NA, (**b**) right NA, (**c**) left amygdala, (**d**) right amygdala, (**e**) frontal medial cortex (FMC)) or for (**f**) task-active voxels across the whole brain. Each dot represents beta coefficients from one voxel estimated with FSL's mixed effects model with outlier deweighting (FLAME 1).

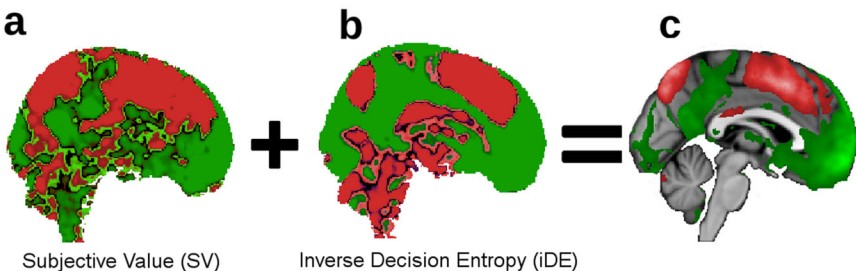

**Fig. 5 Beta weights for subjective value and inverse decision entropy.** For illustration purposes only, we show the gradients that go from dorsal (negative effects in red) to ventral (positive effects in green) in medial prefrontal cortex for (**a**) subjective value, (**b**) inverse decision entropy, and (**c**) summation of inverse decision entropy and subjective value (after z-scoring each variable). Colourless areas in (**c**) represent brain areas where the effects cancel each other out (i.e., close to zero). Lighter areas in (**c**) represent larger absolute values.

relatively uniform gradients of positivity and negativity for both SV and iDE.

## Discussion

The large-scale dataset from the NARPS team afforded us the opportunity to clarify the relationship between SV and a quantity related to confidence, iDE. Previous work by Lebreton et al.[4] suggested that value and confidence combine into a single quantity such that confidence effectively adds to a basic value signal to yield a combined signal that could be used to evaluate actions. This view is supported by data and is intuitive in that being confident in an option should make it more attractive. In addition to the metacognitive roles confidence can play[25,26] in decision making, a combined signal provides an avenue for confidence to impact future choice. Although appealing, this view seems incomplete in that it neglects negative neural coding of confidence—equivalent to presenting stronger activations as confidence diminishes.

We evaluated the possibility that the brain organises value and confidence representations in a systematic fashion that reflects the overall desirability of choice options. This view holds that regions that respond positively to increases in value should also respond positively to increases in confidence. Conversely, there should also be regions that respond negatively to both value and confidence. If the brain represents options in terms of a general notion of desirability that combines value and confidence signals, signals reflecting purely positive and purely negative pairings should be more prevalent than mixed pairings of SV and iDE.

Our view was overwhelmingly supported by the data. As shown in Fig. 3, regions that coded for both SV and iDE tended to code both quantities either positively (e.g., vmPFC) or negatively (e.g., dmPFC). Across the whole brain at the individual voxel level (Fig. 1c), voxels were over-represented that responded positively or negatively to both iDE and SV. This pattern was almost perfectly followed in purported value areas, such as right NA, right amygdala, and frontal medial cortex. Likewise, across voxels, beta weights (and Z statistics) for SV and iDE positively correlated across the whole brain and in purported value areas, particularly in frontal medial cortex (Fig. 4e).

The organisation of positive and negative SV and iDE spans several regions. There appeared to be large gradients in the brain that transition from positive SV and iDE to negative SV and iDE (Fig. 5). Traditional value areas, such as vmPFC, exhibit the positive pairing whereas more dorsal areas display the negative pairing of SV and iDE. In effect, these results complete the satisfying story begun by Lebreton et al.[4]. The tight U-shape relation between SV and iDE is also consistent with studies relating saliency to value[27,28], suggesting further investigation into the relationship between confidence and saliency (see Supplementary Fig. 5 and Median split of SV in the SI).

Our model-based analyses suggests a reinterpretation of purported value areas. Although it was known that confidence signals can appear in purported value areas[6], our results indicate that these confidence signals are stronger and more pervasive in these areas than value signals. This result is striking because these areas were selected because they are understood to be value areas.

One suggestion is that these areas should no longer be referred to as value areas given they are more strongly driven by uncertainty (e.g., iDE) when making risky decisions. Indeed, in this task, there is no strong evidence of pure value signals. Of course, even though these areas are strongly driven by iDE, it would also be incorrect to refer to these areas as uncertainty areas given the intertwined and highly non-accidental relationship between SV and iDE signals. Instead, it appears that decision areas reflect a combined signal that is topographically organised from jointly positive to jointly negative

measures. This suggests that the human brain represents value and confidence along the same spatial axis which could support retrospective evaluation to guide learning and subsequent decisions.

One question is why the brain might organise SV and iDE information in this jointly positive or jointly negative manner. One explanation is that this representation of choice options is easily tied to action[8,29] and is goal-dependent[30]. Such an axis is consistent with valence-dependent confidence[31] and with theories on approach-avoidance being the primary dimension along which behavioural is expressed[32–35]. Studying the role of decision uncertainty in future actions or decisions could help illuminate this link[18,36]. Indeed, evaluating uncertainty negatively is consistent with studies of risk aversion—both in human[7] and non-human primates[19,20]—as well as with intolerance of uncertainty[21]. Thus, our account suggests that confidence and value are integral computations directed toward evaluating action.

Our results support a research strategy of considering how different measures, in this case SV and iDE, relate as opposed to localising single measures. By considering multiple measures and regions, a clear picture emerges of how the brain organises SV and iDE signals, which in turn suggests how this information may be used to support decision making.

This study provides a further lens on the importance of model-based fMRI analyses (for individual participants), which we believe to be more important than issues of method. The model we used was incredibly simple, yet provided the means to understand how SV and iDE signals related. Furthermore, fits to individuals' behaviour yielded measures of loss aversion that reflect individual differences in brain response (see Loss aversion in the brain in the SI). In effect, the cognitive model is demonstrating a reality at both the behavioural and neural level for individual participants, which mirrors recent findings in the concept learning literature on attentional shifts[37,38]. Our results support the claim that cognitive models can reveal intricate facets of behaviour and brain response.

## Methods

**Overview**. Our analyses were based on data from the Neuroimaging Analysis Replication and Prediction Study [NARPS[23,24]]. Data from 108 participants (60 female, 48 male; mean age = 25.5 years, s.d. = 3.59) were made available to participating teams. Informed consent was obtained and the original NARPS study was approved by the ethics committee at Tel Aviv University. The current study was approved by the UCL ethics committee. Participants engaged in a mixed gambles task in an fMRI scanner (four runs). They were asked to either accept or reject gambles based on a 50/50 chance of incurring in a certain amount of monetary gain or loss; where losses and gains were orthogonal to each other. Originally, the available responses were strongly accept, weakly accept, weakly reject, and strongly reject, but these were collapsed into accept and reject categories for our modelling purposes.

Participants were assigned to one of two conditions; an equal range condition and an equal indifference condition. Participants in the equal range condition observed an equal range of potential losses and gains as in ref. [39]. Participants in the equal indifference condition observed a potential range of losses that was half that of potential gains as in ref. [40], consistent with previous estimates of loss aversion (see Experimental protocol and instructions (NARPS) in the SI). Our study did not focus on differences between ranges of gains or losses, thus participants from both conditions were collapsed into a single group. Some participants were previously excluded by the NARPS organisers. We further excluded four participants: one participant had too much head movement (above 2.3 standard deviations above group mean in framewise displacement), one participant reversed the response button mapping, and another two participants were above 2.3 standard deviations from the group mean in either their gain or loss coefficients from our model (see section "Statistics and reproducibility"). Thus, 104 participants were included in the final analyses. Below we summarise our fMRI preprocessing and statistical procedures.

**MRI scanning protocols and fMRI preprocessing**. MRI was performed on a 3T Siemens Prisma scanner at Tel Aviv University. The data were preprocessed by the NARPS organisers using *fMRIPrep* 1.1.6 [[41], RRID:SCR_016216][42]; which is based on *Nipype* 1.1.2[43]; [[44], RRID:SCR_002502]. Brain extraction was performed using the brain mask output from fMRIPrep v1.1.6. (see MRI scanning protocols

(NARPS) and fMRI preprocessing (NARPS) in the SI for more information as well as the information on the NARPS dataset[23,24]).

**Statistics and reproducibility**. For our model-based fMRI analyses, we used SV and iDE as parametric modulators for the general linear model (GLM) of the fMRI data, along with an intercept. This model included temporal derivatives for the mentioned variables and seven movement nuisance regressors (framewise displacement and rotations and translations along the $X$, $Y$, and $Z$ coordinates). The nuisance regressors were all provided as output from fMRIPrep v1.1.6.

Variables in the fMRI GLM were modelled with a double-gamma as a basis function and the full trial duration of four seconds with FSL 5.0.9[45]. No orthogonalisation was forced between regressors but parametric modulators (i.e., SV and iDE) were mean-centred. We used a spatial smoothing kernel of 5mm FWHM and FSL's default highpass filter with 100 seconds cutoff (i.e., locally linear detrending of data and regressors). We also used FSL's default settings for the locally regularised autocorrelation function. The four runs per subject were pooled with fixed effects at the second level and modelled with FSL FEAT's "FLAME 1" with outlier deweighting at the third level.

For inference on the main effects of SV and iDE, we ran whole-brain corrected analyses with FSL's default thresholds for cluster-wise inference of $z = 2.3$ and $p = 0.05$. We looked at both positive and negative activations. To declare activation, or its absence thereof, we took the left and right amygdala, the left and right nucleus accumbens, and the frontal medial cortex masks from the Harvard-Oxford cortical and subcortical atlases provided within FSL. The images were resampled and binarised using FSL's flirt with a threshold of 50%. A custom bash script checked if active voxels were found in these areas as well as doing a visual inspection of the thresholded $z$ maps in the regions of interest. Our choices for the regions of interest (ROI) were based on the original NARPS project; to establish them as a priori decisions. The fact that regions like the nucleus accumbens (NA) or the amygdala show significant correlation between SV and iDE are worthy of notice. Our correlational strategy perhaps is more sensitive to these subtle effects (see Results).

The Results section focused on four different analyses: (1) the negative main effects of SV and iDE, (2) the positive main effects of SV and iDE, (3) the direct comparison of effects between these two variables, and (4) the correlation between SV and iDE across voxels in the brain. For both negative and positive effects, we also reported the results of a conjunction analysis[46] which specifies regions where both variables are significantly below zero (for negative effects, top row in Fig. 3) or above zero (for positive effects, bottom row in Fig. 3). This conjunction analysis was performed as described in ref. [46] using Tom Nichol's *easythresh_conj.sh* script[47].

The third analysis was performed as two one sample $t$-tests with FSL *randomise* (5000 permutations, $p < 0.01$) on the signed differences (i.e., both iDE minus SV and SV minus iDE) between the $Z$ statistics estimated at the second level GLM after pooling estimates with a fixed effects model across the four runs. We use the $Z$ statistics to avoid spurious results based on differences in variance or range between SV and iDE. To account for the fact that a variable can show a larger effect simply because the other variable shows a strong negative effect, we used the conjunction of the contrasts with the corresponding main effects (of either SV or iDE, respectively). To facilitate these conjunctions, we converted the $p$-values from the mentioned FSL *randomise* analysis to $Z$ statistics and further masked the output based on voxels that showed differences in absolute value. Alternatively, testing for differences between absolute values of these variables can be checked in Supplementary Table 3 of the SI. We also report the number of voxels in our cluster activations to emphasise their relative size sampled from MNI152 space at a resolution of 1 mm × 1 mm ×1 mm.

The fourth analysis focuses on the beta weights and the $Z$ statistics (see Correlations of voxelwise $Z$ statistics between SV and iDE in the SI) to compute correlations between SV and iDE across voxels. The voxel activations were estimated across all participants with FSL's FLAME 1 mixed effects model with outlier deweighing and mapped to the MNI template. FSL's mixed effects model considers between-participant variance when estimating activations[48]. Thus, the correlational analysis involved only two statistical maps (one for SV and one for iDE) for the statistic of interest (either beta weights or $Z$ statistics which incorporate voxelwise variance) and after spatial smoothing as detailed above.

**Reporting summary**. Further information on research design is available in the Nature Research Reporting Summary linked to this article.

## Data availability

The original NARPS data can be found at: https://openneuro.org/datasets/ds001734/versions/1.0.4

## Code availability

The code for our main analyses is at: https://github.com/bobaseb/neural_link_SV_iDE

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

## Acknowledgements
The authors thank Benedetto De Martino, Tyler Davis, Rob Mok and Brett Roads for comments on a previous draft of this manuscript. This work was supported by NIH Grant 1P01HD080679, Wellcome Trust Investigator Award WT106931MA, and Royal Society Wolfson Fellowship 183029 to BCL.

## Author contributions
B.C.L. developed the study concept. B.C.L., O.G. and S.B.S. contributed to the study design. O.G. and S.B.S. performed the analysis and interpretation of the behavioural data under the supervision of B.C.L. S.B.S. performed the analysis and interpretation of the fMRI data under the supervision of B.C.L. S.B.S. drafted the manuscript. B.C.L. and O.G. provided critical revisions. All authors approved the final version of the manuscript for submission.

## Competing interests
The authors declare no competing interests.
