## [Peer Review File · Communications Biology]

Reviewers' comments:

Reviewer #1 (Remarks to the Author):

In this manuscript, Bobadilla-Suarez et al aim to investigate the link between the neural encoding of subjective value and confidence. Utilizing a large dataset from the Neuroimaging Analysis Replication and Prediction Study (NARPS), they showed a large extent of overlap of subjective value (SV) and inverse decision entropy (iDE) encoding, especially in the medial PFC.

Overall, there is much to like in this manuscript. It addresses the important issue of decision confidence and its relationship to subjective value, using a novel conceptualization compared to existing studies on this topic. The analysis is clear and straightforward. The large sample size (N=104), which is uncommon in human fMRI studies, is also a major strength of the study. However, there exist a few substantial expositional, conceptual, and methodological concerns that remain to be addressed.

1. The authors may consider reorganizing and tightening the writing of the introduction section, which currently has a wandering style and is, at times, quite confusing. Perhaps most importantly, it needs to flesh out better (a) what exactly is the question this study is trying to address and (b) how this adds to our knowledge of decision neuroscience. I would suggest that more focus be put on a few closely related prior studies (e.g. DeMartino et al., 2013 and Lebreton et al., 2015) and what gaps in those studies are being addressed here.

2. More work needs to be done to demonstrate the validity of operationalizing decision confidence as iDE. To my knowledge, this manuscript is the first study to introduce this construct (a search using the term “inverse decision entropy” on Google Scholar only yields the preprint version of this particular manuscript; a search of “decision entropy” returns a small volume of papers that are mostly outside of neuroscience). A few specific suggestions and questions include:

- a. Rather than talking about iDE as if it was universally known, the authors should try to do a better job in explaining both its precise definition and the underlying intuition, as well as the rationale of adopting iDE as a way to quantify decision confidence, preferably early on in the manuscript.
- b. Another important aspect about iDE that needs clarification is its relationship with SV. The current Figure 1b is somewhat insufficient, and it would be very helpful to complement it by adding a visualization of iDE as a function of SV in its possible range in the experimental paradigm of this study. This could serve two goals: First, it can demonstrate the quantitative relationship between SV and iDE, the two key variables at the center of this study; Second, it also provides empirical evidence that, within the range of SV in this study, these two constructs are sufficiently independent to enable the separate examination of their neural correlates.
- c. Since the authors adopted a novel construct for decision confidence in light of existing studies, it is also critical to more thoroughly discuss and test how it relates to measures of decision confidence used in other studies, for example the outcome variance and subjective confidence ratings. The latter may have particular importance as the current study relies completely on iDS without any validation against how the subjects might feel. In fact, the NARPS dataset does seem to contain some empirical information about their subjective feelings from the available responses of strongly/weakly accept/reject. A comparison of iDE between “strong” and “weak” decisions would be a very useful sanity check to have.

3. An overlooked dimension in the current analyses is valence, given that the space of SV spans both

the positive and negative domains (especially for the “equal range” condition). In fact, a body of existing studies have shown that negative encoding of SV is much more commonly seen in the negative domain, and some have interpreted this as a kind of “saliency” signals (e.g. Litt et al., 2011 Cerebral Cortex, Zhang et al., 2017 Nature Communications). The regions showing negative responses to SV and iDE in this study seem to have substantial overlap with those reported as saliency encoding before. It would be useful to examine if the negative SV and iDE encoding was primarily driven by trials with SVs in the negative domain and, more generally, how SV and confidence encoding might interact with valence.

Intermediate and minor concerns:

1. More details need to be provided regarding the analysis on the correlation between voxel-wise SV beta and iDE beta presented in Figure 4. Is this lumping all voxels and all subjects? If so, this seems to be ignoring potential individual differences and overly liberal. It might be advisable to perform a mixed-effects regression with subject as a random effect. It would be nice to incorporate this in the visualizations (e.g. coloring data points from each subject with a different color) as well.
2. It might be helpful to split Figure 2 into two, one for the 2-by-2 SV and iDE encoding directions and the other for the contrast analysis. A legend is also needed for the coloring scheme.

Reviewer #2 (Remarks to the Author):

In this reanalysis of a public dataset, the authors show that value and confidence signals covary across the brain. This is an important question, given the emphasis given to value in the neuroeconomic field. The idea is that the so-called brain valuation system would not solely encode value, but an aggregation of value and confidence that would represent the overall desirability of an option.

The study has several strengths, the first being a large sample ($n=104$). However, there are also weaknesses that mitigate my enthusiasm.

Major concerns:

- 1) Value and confidence are presented as univocal concepts, but there is no one standard definition. Even in the context of this simple choice task, value and confidence could be computed differently. For instance, subjective value could be computed with curvature parameters on gains and losses, to account for decreasing marginal utility. Confidence has also received many mathematical definitions. For instance, when defined as the subjective probability that the response is correct (as in De Martino’s and Fleming’s studies), confidence is conditional on the choice. Yet another possibility would be to take participants’ responses (‘strong’ versus ‘weak’ yes or no) as a measure of confidence. To generalize the findings, it would be critical to know whether the conclusions hold with these different definitions.
- 2) Comparisons of value and confidence signals rely on a number of voxels in activated clusters or the size of regression estimates. I am afraid these are not valid comparisons, because they depend on the variance of regressors, i.e. on the range of value and confidence levels spanned in the task. So conclusions about a region coding confidence more than value are not grounded. The right analysis to me is the correlation of beta weights across voxels, showing that when a region signals value with increased activity, it also signals confidence with increased activity.

3) A sanity check is missing that value and confidence are truly orthogonal in this design. It is critical because regressors were not orthogonalized in fMRI data analysis, so correlations could arise from biased sampling.

4) The choice of ROI in Figure 3 is surprising. I understand it may come from the original paper, but regions like the amygdala are irrelevant. From Figure 4 it appears that vmPFC and dmPFC are the key payers here, so they are the obvious ROI to investigate.

5) There is a confusion throughout the manuscript between coding low levels of value and confidence, and coding these variables with negative correlation. Otherwise, I do not see why the authors state that previous studies neglected cases where confidence is low, or why risk aversion would be an argument for low confidence coding in the brain. These statements must be clarified.

Minor concerns:

6) The manuscript is poorly written: there are a lot of typos and repetitions, some sentences are not even correct or meaningful. For example (line 209): "The organisation of positive and negative SV and IDE spans regions". Also, some descriptions are vague or loose, for example the weird insistence, even in the abstract, on the fact that maps were smooth. And some statements sound very naïve, such as the final revelation (line 241): "Another general lesson is that model-based fMRI analyses of individual participants are feasible and useful". I reckon the manuscript should be seriously edited to correct both language issues and scientific approximations.

7) Figure legends are unclear. In figure 1, I could not make sense of the lotteries placed in the different cells of the design. In figure 3, I guessed that dots represent activity in voxels of a brain template (averaged across participants). This should be specified.

8) Methods are not sufficiently detailed. For example, how maps were thresholded, or how correlations across voxels were assessed and tested statistically (before or after smoothing?) should be explained.

9) The investigation of loss aversion presented as supplementary information is interesting but disconnected from the main question. I suggest to remove it.

Referee expertise:

Referee #1: neuroeconomics, game-theory, decision-making

Referee #2: decision making, neuropsychology, computational modeling

Reviewers' comments:

Reviewer #1 (Remarks to the Author):

In this manuscript, Bobadilla-Suarez et al aim to investigate the link between the neural encoding of subjective value and confidence. Utilizing a large dataset from the Neuroimaging Analysis Replication and Prediction Study (NARPS), they showed a large extent of overlap of subjective value (SV) and inverse decision entropy (iDE) encoding, especially in the medial PFC.

Overall, there is much to like in this manuscript. It addresses the important issue of decision confidence and its relationship to subjective value, using a novel conceptualization compared to existing studies on this topic. The analysis is clear and straightforward. The large sample size (N=104), which is uncommon in human fMRI studies, is also a major strength of the study. However, there exist a few substantial expositional, conceptual, and methodological concerns that remain to be addressed.

Thank you for your effort and the constructive feedback.

1. The authors may consider reorganizing and tightening the writing of the introduction section, which currently has a wandering style and is, at times, quite confusing.

We appreciate this suggestion and have now rewritten substantial parts of the Introduction to be tighter and clearer than before. See below for more specific information on the changes made.

Perhaps most importantly, it needs to flesh out better (a) what exactly is the question this study is trying to address

To more succinctly state this study's aim we have added the following sentence to the third to last paragraph in the Introduction (line 61):

“Thus, this study characterizes the joint neural coding of value and confidence on the medial surface of the human brain.”

and (b) how this adds to our knowledge of decision neuroscience.

We explicitly state the knowledge contribution to decision neuroscience in the second to last paragraph of the Introduction (lines 66-70):

“The contribution of this study to decision neuroscience is twofold. First, the joint coding of value and confidence, previously proposed by Lebreton et al . (2015) for positive value and high confidence, is extended to consider the converse case and the distribution of voxels jointly coding value and confidence. Furthermore, our results suggest that medial surface activity is best described by large scale maps for decision and action related computations.”

I would suggest that more focus be put on a few closely related prior studies (e.g. DeMartino et al., 2013 and Lebreton et al., 2015) and what gaps in those studies are being addressed here.

We thank the reviewer for this constructive suggestion. We have now rewritten the Introduction to be more concise and focus on relevant previous work by Dr. Maël Lebreton (2015) and Dr. Benedetto De Martino (2013). We noticed that the Introduction already focused on the mentioned studies so we removed paragraphs (see note on line 36) that detracted from that focus. We believe this more succinct Introduction better communicates our objectives. We have also defined some of our terms more clearly, like what we mean by smooth maps (as suggested by R2).

2. More work needs to be done to demonstrate the validity of operationalizing decision confidence as iDE.

As mentioned above, we examined the relationship between the different levels of participant response (weakly reject and weakly accept vs. strongly reject and strongly accept) with inverse decision entropy (iDE). We found a significant difference between the weak and strong levels on P. 4 (lines 112-118):

“To evaluate the robustness of iDE, we considered how it varied for strongly vs. weakly accepts and rejects. Although we modeled iDE based on the accept vs. reject binary distinction, participants had four responses available to them. Even though our model fit was not informed by the strongly vs. weakly distinction, one would hope that iDE would be lower for the weakly accept and reject responses than for the strongly accept and reject responses. Indeed, as shown in Figure 2b, this relation held. The lower confidence responses (Weakly Reject and Weakly Accept) showed lower iDE (mean Weak iDE of -0.55, s.d. = 0.189) than the Strongly Accept and the Strongly Reject options (mean Strong iDE of -0.20, s.d. = 0.177, significantly higher than Weak iDE: $t(406.61) = 19.31, p < 0.001$).”

We also include Figure 2 (right panel) to help visualise this result. This new analysis helps validate intuitions underlying the measure.

Figure 2: (a) The U-shaped relationship between p_{accept} and iDE is shown for over twenty-four thousand choices across all participants ($N = 104$). Confidence is highest for low-value gambles that have a low probability of acceptance and high-value gambles that have a high probability of acceptance. The transparency in the plot reflects the density of observations in the empirical data along the p_{accept} horizontal axis. (b) A plot of iDE as a function of the four possible responses: Strongly Reject, Weakly Reject, Weakly Accept, Strongly Accept. Each dot is a participant’s mean for that response type and the grey conveys the density across participants. Although our cognitive model was fit to the binary distinction of accept vs. reject, it successfully generalized by showing sensitivity to the weakly vs. strong distinction for which it was not fit.

Furthermore, we looked at data from one of the experiments in a study published in *Nature Human Behavior* (Folke et al., 2017). We include this information in the Supplemental Information (lines 209-240 of the SI, P. S7, section titled “Validation of iDE”). We mention in the main text (P. 4, lines 105-111) that iDE correlates both with explicit confidence ratings – from the study by Folke and colleagues (2017) – as with the subjective probability of being

correct, which has also been used as an operationalisation of confidence in previous studies (De Martino et al., 2013):

“Our iDE measure of confidence closely tracks other measures in the literature. For example, iDE positively correlates with confidence ratings provided by participants in a behavioral study ($n = 28$, Folke et al., 2017, see SI) of value-based decision making with a Spearman correlation of 0.45 (s.d. = 0.171, $t(27) = 13.61$, $p < 0.001$). In that study, iDE was closely related to the authors' preferred definition of confidence, namely the subjective probability of being correct (above zero mean Spearman correlation of 0.89, s.d. = 0.114, $t(27) = 40.65$, $p < 0.001$). This measure of confidence and iDE also tracked one another in the current study using the NARPS data (above zero mean Spearman correlation of 0.96, s.d. = 0.046, $t(103) = 210.426$, $p < 0.001$). These relations hold for alternative definitions of value as well (see SI).”

We believe these results demonstrate the validity of operationalising decision confidence as iDE.

To my knowledge, this manuscript is the first study to introduce this construct (a search using the term “inverse decision entropy” on Google Scholar only yields the preprint version of this particular manuscript; a search of “decision entropy” returns a small volume of papers that are mostly outside of neuroscience). A few specific suggestions and questions include:

a. Rather than talking about iDE as if it was universally known, the authors should try to do a better job in explaining both its precise definition and the underlying intuition, as well as the rationale of adopting iDE as a way to quantify decision confidence, preferably early on in the manuscript.

We agree with the reviewer that the concept of inverse decision entropy deserves a gentler introduction. To that purpose, we have now added these lines to the beginning of the second paragraph in the Introduction (lines 17-22):

“Shannon entropy is a well-formulated measure of uncertainty (Shannon, 1948) that is well suited for examining confidence. So that it positively aligns with confidence, we consider the inverse of the entropy associated with a person's decision, which we refer to as inverse decision entropy (iDE). Shannon entropy characterises the uncertainty for a probability distribution in terms of the expected self-information, which can be calculated as the sum

of the probability of each state times its log probability. In the case of the binary decisions considered here, the probability distribution is simply a binomial.”

Along with the new figures and validations described below, the concept should now be easier for readers to follow.

b. Another important aspect about iDE that needs clarification is its relationship with SV. The current Figure 1b is somewhat insufficient, and it would be very helpful to complement it by adding a visualization of iDE as a function of SV in its possible range in the experimental paradigm of this study. This could serve two goals: First, it can demonstrate the quantitative relationship between SV and iDE, the two key variables at the center of this study; Second, it also provides empirical evidence that, within the range of SV in this study, these two constructs are sufficiently independent to enable the separate examination of their neural correlates.

Indeed the relationship between SV and iDE could be made more explicit. We note that SV doesn't align well across subjects due to differences in loss aversion and indifference points. Interestingly, this individual difference that is assessed through model fits to behaviour has an observable neural signature (see the section on “Loss aversion in the brain” in the SI). Because of these individual differences, for visual presentation and without loss of generality, we plotted p_{accept} against iDE (left panel of Figure 2). This panel further shows transparency-coded information about the density of p_{accept} for all participants' choices (approximately twenty-four thousand choices in total). The variable p_{accept} is a monotonic transformation of SV; with a mean Spearman correlation of 0.99 (s.d. = 0.052) between variables – practically no information is lost. This information is reported on P. 4 (lines 98-104):

“As can be seen in Figure 2a, iDE has a quadratic relation to p_{accept} with a significant (above zero) mean Spearman correlation of 0.16 (s.d. = 0.568, $t(103) = 2.88$, $p = 0.005$) across participants. The density of observations for p_{accept} – estimated with one thousand bins for over twenty-four thousand choices across participants – is biased towards towards the upper and lower bounds (i.e., $p_{\text{accept}} = 1$ and $p_{\text{accept}} = 0$, respectively). Likewise, iDE shares a quadratic relation with SV (see SI) presenting a significant (above zero) mean Spearman correlation of 0.16 (s.d. = 0.568, $t(103) = 2.88$, $p = 0.005$), which follows from the high Spearman correlation between SV and p_{accept} (mean = 0.99, s.d. = 0.0004, one sample t -test above zero: $t(103) = 28687.295$, $p < 0.001$). ”

For completeness, we include the individual plots for each participant between SV and iDE in the SI (Figure S3, P. S9):

Figure S3: Scatterplots of iDE by SV per participant. Each panel represents a participant (104 in total). The y-axis presents inverse decision entropy (iDE) and the x-axis presents (z-scored) subjective value (SV). Each dot represents one choice for a given gamble.

c. Since the authors adopted a novel construct for decision confidence in light of existing studies, it is also critical to more thoroughly discuss and test how it relates to measures of decision confidence used in other studies, for example the outcome variance and subjective confidence ratings. The latter may have particular importance as the current study relies completely on iDS without any validation against how the subjects might feel. In fact, the NARPS dataset does seem to contain some empirical information about their subjective feelings from the available responses of strongly/weakly accept/reject. A comparison of iDE between “strong” and “weak” decisions would be a very useful sanity check to have.

As mentioned above, these suggestions were on point and we have validated our iDE measure both with levels of participant response (right panel of Figure 2) and with alternative measures of confidence: both explicit confidence ratings and $p_{correct}$ (subjective probability of being correct) are addressed in the Results section (P. 4, lines 105-111) and in the SI (lines 209-240 on P. S7-S8 and Figure S2 in the SI).

Figure S2: Spearman correlation matrices for NARPS and a two alternative forced choice task from Folke et al. (2017). Left panel shows the (Spearman) correlation matrix for several variables estimated from the NARPS behavioral data: subjective value (SV), p_{accept} , inverse decision entropy (iDE), and p_{correct} , which is the subjective probability of being correct. The middle and right panels are based on behavioral data from Folke et al. (2017) for a similar set of variables: difference in value (DV), iDE, confidence (i.e., explicit ratings), and p_{correct} . The middle panel estimates all variables (except for the confidence ratings) based on explicit reports of willingness-to-pay values for the choice options (i.e., snacks) from a Becker–DeGroot–Marschak (BDM) auction. However, the right panel does not use such BDM values, but estimates them directly from the choices instead. DV is simply the subjective value of an item presented on the right minus the value of an item presented on the left in that experiment.

3. An overlooked dimension in the current analyses is valence, given that the space of SV spans both the positive and negative domains (especially for the “equal range” condition). In fact, a body of existing studies have shown that negative encoding of SV is much more commonly seen in the negative domain, and some have interpreted this as a kind of “saliency” signals (e.g. Litt et al., 2011 Cerebral Cortex, Zhang et al., 2017 Nature Communications). The regions showing negative responses to SV and iDE in this study seem to have substantial overlap with those reported as saliency encoding before. It would be useful to examine if the negative SV and iDE encoding was primarily driven by trials with SVs in the negative domain and, more generally, how SV and confidence encoding might interact with valence.

Interestingly, the original NARPS project focused on hypotheses related to gains and losses, though our focus is slightly different which prompted us to work with a combined subjective value term that reflects how gains and losses are combined. To provide another vantage point on our results, we performed a median split on SV to create two variables for which we could examine positive and negative effects. We report this exact analysis in the SI (lines 349-366 on P. S17 and Figure S5; section titled “Median split of SV”), where we include SV below and above the participant specific median as two separate variables in the general linear model (GLM) along with iDE. The results are as we would expect with the positive variable patterning with iDE as reported in the main text and the negative

variable showing the converse pattern. These results are consistent with the notion that iDE is capturing neural effects related to distance from the indifference point of the decision.

Figure S5: Positive and negative activations of median split SV and iDE on the medial surface. The left and middle columns show activations for SV above the participant specific median and below the participant specific median, respectively. The right column shows activations for iDE. The top row presents positive activations and the bottom row presents negative activations.

To reiterate the points above, the interpretation of splitting SV around participant specific medians (effectively near each participant's indifference point) should be handled with care. Since SV's distance to the indifference point is the main phenomenon that iDE attempts to capture, it is expected that there would be a crossover interaction between the relation between iDE and SV below the median and iDE with SV above the median as shown in the SI (lines 349-366 on P. S17, section titled "Median split of SV"). The correlation is positive in the latter but negative in the former (see P. S17-S18 of the SI). This analysis further supports our main premise that iDE is a more fundamental quantity in the medial surface than SV.

Intermediate and minor concerns:

- 1. More details need to be provided regarding the analysis on the correlation between voxel-wise SV beta and iDE beta presented in Figure 4. Is this lumping all voxels and all subjects? If so, this seems to be ignoring potential individual differences and overly liberal. It might be advisable to perform a mixed-effects regression with subject as a*

random effect. It would be nice to incorporate this in the visualizations (e.g. coloring data points from each subject with a different color) as well.

The details regarding this analysis have now been further clarified in the manuscript (caption of what is now Figure 4 and in the subsection titled “Interdependence of subjective value and inverse decision entropy” in the Results section, lines 194-200). A statistical map of beta coefficients was estimated across participants for SV and iDE using FSL’s mixed effects model with outlier deweighting. It is the correlations between these two maps, for certain regions of interest, that are presented. Since these estimates were a result of FSL’s mixed effects algorithm (FLAME), they already integrate between-participant variance in the estimate. We now include a reference (Woolrich et al., 2004) to this in the manuscript. This is unlike SPM’s default behavior which assumes homogenous subject variance (see Mumford & Nichols, 2009, for more details). To further address this concern, we now report the correlation of the Z statistics between both variables in the SI. This statistic integrates voxel-specific variance across participants and effectively presents slightly lower correlations than the beta coefficients as expected (section titled “Correlations of voxelwise Z statistics between subjective value and inverse decision entropy”; lines 335-348 on P. S17 of the SI).

2. It might be helpful to split Figure 2 into two, one for the 2-by-2 SV and iDE encoding directions and the other for the contrast analysis. A legend is also needed for the coloring scheme.

We appreciate the suggestion made by the reviewer but we prefer to present the contrasts along with the main effects since it is part of a central point we intend to make; iDE is effectively dominating activations on the medial surface when compared to SV. A legend has now been added for the colouring scheme (now Figure 3).

Figure 3: Main effects and contrasts in medial prefrontal cortex. Presents significant activations (Z statistical maps) of subjective value (purple), inverse decision entropy (dark pink), and their conjunction (light pink) for a whole-brain corrected analysis conducted with FSL FEAT's FLAME 1 for different combinations of positive and negative main effects (2x2). The column on the right hand side (i.e., **contrasts**) shows areas with stronger negative effects (top right) or stronger positive effects (lower right).

Reviewer #2 (Remarks to the Author):

In this reanalysis of a public dataset, the authors show that value and confidence signals covary across the brain. This is an important question, given the emphasis given to value in the neuroeconomic field. The idea is that the so-called brain valuation system would not solely encode value, but an aggregation of value and confidence that would represent the overall desirability of an option.

The study has several strengths, the first being a large sample (n=104). However, there are also weaknesses that mitigate my enthusiasm.

Thank you for providing a helpful review that helped us improve the manuscript.

Major concerns:

1) Value and confidence are presented as univocal concepts, but there is no one standard definition. Even in the context of this simple choice task,

value and confidence could be computed differently. For instance, subjective value could be computed with curvature parameters on gains and losses, to account for decreasing marginal utility. Confidence has also received many mathematical definitions. For instance, when defined as the subjective probability that the response is correct (as in De Martino's and Fleming's studies), confidence is conditional on the choice. Yet another possibility would be to take participants' responses ('strong' versus 'weak' yes or no) as a measure of confidence. To generalize the findings, it would be critical to know whether the conclusions hold with these different definitions.

The reviewer is correct in highlighting the diversity of operationalisations for value and confidence. We address this both in the SI (lines 209-240 on P. S7-S8) and in the main text (lines 105-118 on P. 4). In the SI we explore in detail alternative operationalisations of value and confidence. As mentioned above, we relate iDE to levels of participant response for the current study but we also relate it to explicit confidence ratings and the subjective probability of being correct. These are all analogous ways of operationalising the construct of interest (i.e., confidence). Furthermore, alternative views on value are also explored. The data from Folke and colleagues (2017) is based on a two alternative forced decision task between retail items (i.e., snacks). In contrast to the data from this study, the value of each item is not provided by the experimenter as in the mixed gambles but elicited from the participants both through their choices and through a Becker–DeGroot–Marschak (BDM) auction. For each choice, it is normal practice to estimate the difference in (subjective) value between the items with the monetary values elicited in the BDM auction. We also estimate these directly from participant choices and show that these estimates strongly correlate with those from the BDM auction.

Thus we now include a section titled “Validation of iDE” in the SI (lines 209-240 on P. S7-S8). There we present three different correlation matrices with variants of operationalisations of both value and confidence, including the suggested subjective probability of being correct which is conditional on choice (i.e., $p_{correct}$). The right panel of Figure 2 further demonstrates the validity of iDE, showing its quadratic relation to the four levels of participant response. The respective average Spearman correlations are presented in the Results section (lines 98-111 on P. 4). Thank you for prompting us to do this work that provides a more solid foundation for our findings.

2) Comparisons of value and confidence signals rely on a number of voxels in activated clusters or the size of regression estimates. I am afraid these are not valid comparisons, because they depend on the variance of

regressors, i.e. on the range of value and confidence levels spanned in the task. So conclusions about a region coding confidence more than value are not grounded.

This is a point we took considerable precaution with in the initial submission and we now further elaborate on our logic in the Methods section (lines 320-321 on P. 10). For each participant, we estimate Z maps (via a transformation of the p values for each voxel). We compare SV and iDE using FSL's *randomise* permutation function (5000 iterations). The use of Z maps, as opposed to beta coefficients, should deal with the concern that the scale and range of the original variables are not comparable. Given that direct comparisons between number of activated voxels is uncommon in the literature, we have removed that section from the Discussion (see note on line 235).

The right analysis to me is the correlation of beta weights across voxels, showing that when a region signals value with increased activity, it also signals confidence with increased activity.

We agree with the reviewer that the correlations between maps is a sensible next step, which was also our reasoning for the initial submission. We now include the correlations for the final Z maps between SV and iDE that incorporate voxel-specific variance across participants (lines 335-348 on P. S17 of the SI). These correlations, though slightly lower than their counterpart in the analysis with the beta coefficients, remain strong and robust as expected. As mentioned above, both the betas and the Z statistics are estimated from FSL's mixed effects model with outlier deweighting (i.e., FLAME 1) which inherently considers between-participant variance (Woolrich et al., 2004).

3) A sanity check is missing that value and confidence are truly orthogonal in this design. It is critical because regressors were not orthogonalized in fMRI data analysis, so correlations could arise from biased sampling.

The referee makes a good suggestion for doing sanity checks of the correlations between confidence and value as well as with their alternative operationalisations. Usually orthogonalisation is not advised (Mumford, Poline & Poldrack, 2015). However, we agree that assessing correlations between regressors to avoid collinearity, as well as mean-centering parametric modulators (as we do), is good practice. We present such correlation matrices in Figure S2 of the SI. We show that iDE does not correlate with difference in value (DV) in the Folke et al. (2017) data and only correlates mildly for the current study as it does for alternative definitions of confidence (i.e., the subjective probability of being

correct). The details are now included both in the SI (section titled “Validation of iDE”; lines 209-240 on P. S7-S8 of the SI) and in the Results section (lines 105-111): 1) a mild average Spearman correlation is found (mean $r = 0.17$), which does not raise serious concerns regarding biased sampling or issues of collinearity for the GLM, 2) this correlation is the same for p_{accept} , further establishing its close link with SV, and 3) a similarly mild correlation between SV and $p_{correct}$ is also reported in the SI (mean $r = 0.16$).

4) The choice of ROI in Figure 3 is surprising. I understand it may come from the original paper, but regions like the amygdala are irrelevant. From Figure 4 it appears that vmPFC and dmPFC are the key players here, so they are the obvious ROI to investigate.

Indeed, our choices for the regions of interest (ROI) were based on the original NARPS project, which considered areas believed to be related to value. We embarked on this path to avoid criticisms in how we chose the ROIs; allowing us to establish them as *a priori* decisions. The fact that regions like the nucleus accumbens (NA) or the amygdala show significant correlation between SV and iDE is worthy of notice. Our correlational strategy may be more sensitive to these subtle effects and we would prefer to leave their presentation in the main text (now explicitly stated in the section titled “Model-based fMRI” in the Methods on lines 308-309).

5) There is a confusion throughout the manuscript between coding low levels of value and confidence, and coding these variables with negative correlation. Otherwise, I do not see why the authors state that previous studies neglected cases where confidence is low, or why risk aversion would be an argument for low confidence coding in the brain. These statements must be clarified.

We did not intend to suggest that low levels of value and confidence have been neglected in past studies. In the revision, we have taken care to separate the distinction between neural coding of a variable (with negative or positive activations) and low or high levels of such variables. Specifically, we have modified a section in the fourth to last paragraph in the Introduction (lines 50-54) to:

“We find this basic account appealing, but incomplete. Lebreton et al. (2015) focused on the case of positive coding of value and confidence in vmPFC. If value and confidence signals are truly intertwined, then there should also be regions that code the converse; negative coefficients for value and confidence, which is equivalent to increased activity for low confidence and negative value.”

We hope this clarification in the Introduction will guide the reader on what we mean by negative neural coding (i.e., negative beta coefficients) as being the same as stronger activations for lower values.

Minor concerns:

6) The manuscript is poorly written: there are a lot of typos and repetitions, some sentences are not even correct or meaningful. For example (line 209): “The organisation of positive and negative SV and IDE spans regions”. Also, some descriptions are vague or loose, for example the weird insistence, even in the abstract, on the fact that maps were smooth. And some statements sound very naïve, such as the final revelation (line 241): “Another general lesson is that model-based fMRI analyses of individual participants are feasible and useful”. I reckon the manuscript should be seriously edited to correct both language issues and scientific approximations.

We have gone through the entire manuscript to improve readability and diminish repetitions throughout. Line 227 (previously line 209) has now been changed to “The organisation of positive and negative SV and IDE spans several regions”. Line 255 (previously line 241) has now been modified as “This study provides a further lens on the importance of model-based fMRI analyses (for individual participants), which we believe to be more important than issues of method.” We also reduce the usage of “smooth maps” where appropriate. We now provide two references (Guest & Love, 2017, Margulies et al., 2016; lines 56-57 in the third to last paragraph in the Introduction) and further clarify what we mean by smooth maps in the last paragraph of the Results (lines 201-202):

“Third, the relation between SV and iDE formed smooth maps, as opposed to parcellations (Margulies et al., 2016), that spanned large regions that were either positive or negative for both SV and iDE.”

The literature on brain gradients (synonymous with smooth maps) is quickly expanding as showcased by a recent virtual conference organised by the Montreal Neurological Institute (MNI). Our framing is consistent with another value gradient we described on the medial surface in a previous study (De Martino, Bobadilla-Suarez, Noguchi, Sharot, & Love, 2016; referenced on lines 33 and 41). The implications of smoothness of the BOLD signal for the neural code is further discussed in Guest and Love (2017), which we also now reference in the Introduction (line 56).

7) Figure legends are unclear. In figure 1, I could not make sense of the lotteries placed in the different cells of the design. In figure 3, I guessed that dots represent activity in voxels of a brain template (averaged across participants). This should be specified.

We thank the reviewer for pointing this out. The captions for Figure 1 and Figure 3 (now Figure 4) have now been extended. We also detail our correlational analysis further in the Results section (lines 194-200, first paragraph on P. 8).

8) Methods are not sufficiently detailed. For example, how maps were thresholded, or how correlations across voxels were assessed and tested statistically (before or after smoothing?) should be explained.

The fourth to last paragraph of the Methods describe the thresholding used to declare activation in a region (lines 300-306). We have a more detailed explanation of the spatial voxel correlations (last paragraph of the Methods section, lines 328-333). We also added that the voxel correlations were performed after spatial smoothing (line 333).

9) The investigation of loss aversion presented as supplementary information is interesting but disconnected from the main question. I suggest to remove it.

The section on loss aversion in the SI serves the function of further validating our behavioural model. It is important to show that a relationship exists between losses and gains in the brain and at the behavioural level. Please notice the issue of individual differences arises in response to Reviewer 1, which ties into this finding. We think these fundamental analyses justify our usage of SV as well. To our knowledge, this is a novel way of relating behavioural loss aversion to neural loss aversion and we would prefer to keep the finding in the SI because we believe many readers in this area will appreciate the finding. We have added a few sentences to that section to better motivate the analysis.

REVIEWERS' COMMENTS:

Reviewer #1 (Remarks to the Author):

This submission by Bobidilla-Suarez et al is the revised version of an earlier manuscript I reviewed a while ago. In this study, the authors aim to investigate the link between the neural encoding of subjective value and confidence. Utilizing a large public dataset from the NARPS, they showed a large extent of overlap of subjective value (SV) and inverse decision entropy (iDE) encoding, especially in the medial PFC.

Overall, the authors have been responsive to my and the other reviewer's comments. In particular, the additional results included on the validity of iDE as a measure of confidence and its consistency with other confidence measures in both this dataset and in previous studies are especially helpful. I still have the following concerns related to some of my other comments last time:

1. While the introduction section has been improved in this revision, it still does not seem to deliver a clear message on what this paper is promising to deliver on, or why this question is of significance. The abstract suffers from the similar problem, perhaps to a larger extent, because it is exclusively a summary of the hypothesis and the results and remains completely silent on the big-picture question and its significance. More specifically, the research question is not sufficiently motivated. Is this study a mere extension of Lebreton et al (2015) or something more? The transition to the prediction about gradient-like cortical maps (line 56) is also abrupt and seems to come out of nowhere. Most importantly, there is essentially no discussion on the "so what" question – given the topic is clearly behaviorally-motivated, what insights would we gain on behavior from this study? This should be clearly spelled out in the introduction independent of the actual results obtained, but even in the discussion section at the end of the article, the authors sound quite vague on this front.

I am aware that this study is a re-analysis of a public dataset, but I don't think the standard of a crystal clear description of the research question and its significance should be any lower just because the experimental design and data collection were not under the authors' control. I also believe that a bit more effort on this would be worth the return of increased impact of the paper.

2. The references to the figures follow a peculiar order that are almost entirely different from the order of the figures themselves, which almost certainly creates confusion and inconvenience to the readers. The order of the figures being mentioned in the text is 2a, 1b, 1c, 1a, 2b, 3, 1c again, 4, and 5. I honestly don't understand why such hopping around is necessary.

3. On my previous point #3 on valence, the additional analysis in the Supplement "Median split of SV" that the authors include in this revision is helpful. I hope that the authors could discuss in the manuscript the relationship between the confidence encoding in this study vs. the similarly U-shaped saliency encoding in those studies. At face value, they are very similar despite completely different theoretical underpinnings. If this study does not allow one to distinguish the two accounts, it might be helpful to acknowledge this caveat. If the authors believe that there is reason to think otherwise, they should explain it more explicitly.

Reviewer #2 (Remarks to the Author):

The authors have addressed all my concerns and I am globally happy with the revisions.

Some presentational concerns are remaining, but these might be fixed at a later stage with editors / publishers.

For instance, in Fig. 1C, the lotteries provided as examples are not helpful, because we need the beta weights for gains and losses to estimate subjective value and inverse entropy. These estimates should be made explicit on the figure.

Also, there are remaining typos, e.g. legend in Fig. 3 starts a sentence with "Presents significant activation ...". Besides, this figure is hard to grasp. It took me a while to understand that contrasts were not about comparing the two variables (SV versus IDE). For completeness (looking for differences and not just commonalities), this contrast between the two variables of interest should also be shown.

REVIEWERS' COMMENTS:

Reviewer #1 (Remarks to the Author):

This submission by Bobadilla-Suarez et al is the revised version of an earlier manuscript I reviewed a while ago. In this study, the authors aim to investigate the link between the neural encoding of subjective value and confidence. Utilizing a large public dataset from the NARPS, they showed a large extent of overlap of subjective value (SV) and inverse decision entropy (iDE) encoding, especially in the medial PFC.

We thank the reviewer for their help throughout this process.

Overall, the authors have been responsive to my and the other reviewer's comments. In particular, the additional results included on the validity of iDE as a measure of confidence and its consistency with other confidence measures in both this dataset and in previous studies are especially helpful. I still have the following concerns related to some of my other comments last time:

1. While the introduction section has been improved in this revision, it still does not seem to deliver a clear message on what this paper is promising to deliver on, or why this question is of significance.

To this end, we have added the following sentence to the penultimate paragraph of the Introduction (lines 69-71):

“Our results indicate that subjective value and confidence gradients in the brain are aligned in a manner that reflects the overall desirability of a decision, which could be useful in retrospective evaluation of a decision.”

The abstract suffers from the similar problem, perhaps to a larger extent, because it is exclusively a summary of the hypothesis and the results and remains completely silent on the big-picture question and its significance.

The Abstract (lines 1-11) now mentions the behavioural interest of seeing value and confidence jointly encoded in aligned gradients as well as the potential utility for retrospective evaluation (making this work's significance clearer):

“Recent work has considered the relationship between value and confidence in both behavior and neural representation. Here we evaluated whether the brain organizes value and confidence signals in a systematic fashion that reflects the overall desirability of options. If so, regions that respond to either increases or decreases in both value and confidence should be widespread. We strongly confirmed these predictions through a model-based fMRI analysis of a mixed gambles task that assessed subjective value (SV) and inverse decision entropy (iDE), which is related to confidence. Purported value areas more strongly signalled iDE than SV, underscoring how intertwined value and confidence are. A gradient tied to the desirability of actions transitioned from positive SV and iDE in ventromedial prefrontal cortex to negative SV and iDE in dorsal medial prefrontal cortex. This alignment of SV and iDE signals could support retrospective evaluation to guide learning and subsequent decisions.”

More specifically, the research question is not sufficiently motivated. Is this study a mere extension of Lebreton et al (2015) or something more?

Although we build on work from Lebreton, Fleming, and De Martino, our results suggest that the human brain organises value and confidence along the same spatial axis in the brain and that confidence is more strongly represented than value. Speculatively, this could be useful for functions such as retrospective evaluation of decisions. We clearly state this at the end of the Introduction as mentioned above and restate in the sixth paragraph of the Discussion (lines 246-247):

“This suggests that the human brain represents value and confidence along the same spatial axis which could support retrospective evaluation to guide learning and subsequent decisions.”

The transition to the prediction about gradient-like cortical maps (line 56) is also abrupt and seems to come out of nowhere.

We have added that our focus is on brain gradients (as opposed to clear-cut localizations) along with the citation of Margulies et al. (2016) at the end of the fourth paragraph of the Introduction (lines 42-44):

“At this juncture, rather than focusing on their localization, we suggest mapping the relationship between confidence and value throughout the brain with a focus on gradients (Margulies et al., 2016).”

This should setup its mention further below (previously line 56).

Most importantly, there is essentially no discussion on the “so what” question – given the topic is clearly behaviorally-motivated, what insights would we gain on behavior from this study? This should be clearly spelled out in the introduction independent of the actual results obtained, but even in the discussion section at the end of the article, the authors sound quite vague on this front.

We thank the reviewer for pushing us to emphasize the big picture relevance of our study. We believe its motivation and relation to behaviour is now clearly stated in the additions mentioned above. Its potential utility for retrospective evaluation provides a clear advantage for future behaviour.

I am aware that this study is a re-analysis of a public dataset, but I don't think the standard of a crystal clear description of the research question and its significance should be any lower just because the experimental design and data collection were not under the authors' control. I also believe that a bit more effort on this would be worth the return of increased impact of the paper.

We agree, the standards should not differ based on the reanalysis of previous data. We thank the reviewer again for their input which we believe has greatly improved our message. The changes to the Abstract, Introduction and Discussion have greatly helped on this front.

2. The references to the figures follow a peculiar order that are almost entirely different from the order of the figures themselves, which almost certainly creates confusion and inconvenience to the readers. The order of the figures being mentioned in the text is 2a, 1b, 1c, 1a, 2b, 3, 1c again, 4, and 5. I honestly don't understand why such hopping around is necessary.

We can see how this can be confusing to readers and thank the reviewer for this observation. We have now adjusted figure mentions to follow the

correct order of presentation (up until the Discussion where referring back is necessary) with the exception of the mention of Figure 1c in the third section of the Results.

3. On my previous point #3 on valence, the additional analysis in the Supplement "Median split of SV" that the authors include in this revision is helpful. I hope that the authors could discuss in the manuscript the relationship between the confidence encoding in this study vs. the similarly U-shaped saliency encoding in those studies. At face value, they are very similar despite completely different theoretical underpinnings. If this study does not allow one to distinguish the two accounts, it might be helpful to acknowledge this caveat. If the authors believe that there is reason to think otherwise, they should explain it more explicitly.

The reviewer is insightful in directing our attention to this connection which we have now discussed at the end of the fourth paragraph of the Discussion (lines 234-236) with relevant citations and pointers to the section in the supplemental:

"The tight U-shape relation between SV and iDE is also consistent with studies relating saliency to value (Litt et al., 2011; Zhang et al., 2017), suggesting further investigation into the relationship between confidence and saliency (see Median split of SV in the SI)."

Reviewer #2 (Remarks to the Author):

The authors have addressed all my concerns and I am globally happy with the revisions.

We are glad the reviewer is satisfied with our revisions and thank them for their help throughout this process.

Some presentational concerns are remaining, but these might be fixed at a later stage with editors / publishers.

For instance, in Fig. 1C, the lotteries provided as examples are not helpful, because we need the beta weights for gains and losses to estimate subjective value and inverse entropy. These estimates should be made explicit on the figure.

We thank the reviewer for making this keen observation. We have now included the weight ratio between gains in losses at the end of the caption of Figure 1.

*Also, there are remaining typos, e.g. legend in Fig. 3 starts a sentence with "Presents significant activation ..."
Besides, this figure is hard to grasp. It took we a while to understand that contrasts were not about comparing the two variables (SV versus IDE). For completeness (looking for differences and not just commonalities), this contrast between the two variables of interest should also be shown.*

The typo has been fixed and the reviewer was in fact correct that the contrasts were about comparisons between SV and IDE. This has now been explicitly stated in the caption of Figure 3.